# OpenReview forum: "Accumulating Context Changes the Beliefs of Language Models"
_ICLR.cc/2026/Conference — Submitted to ICLR 2026_

### Official Review · Reviewer_dTDQ · 2025-10-29

**Soundness:** 2
**Presentation:** 4
**Contribution:** 3
**Rating:** 4
**Confidence:** 3

**Summary:**

This paper studies how accumulated context in LM assistants can influence their responses and behavior over time. The authors propose a three-stage framework to measure changes in both stated beliefs and action after the model engages in rounds of  debate, persuasion, reading, or research. Experiments on multiple models, including GPT-5 and Grok 4, show that the conditional output distributions are highly sensitive to prior context.

**Strengths:**

1. This paper identifies belief shift as a core risk of long-context LM behavior.


2. Experiments mimic real usage scenarios (debate, reading, research).


3. Empirical results reveal divergence between stated beliefs and actual behavior, which is critical for agent alignment.

**Weaknesses:**

1. The paper appears to rebrand statistical shifts in output probabilities as ''belief change.'' Here, ''belief'' refers to the model’s response distribution conditioned on accumulated context, which seems to differ a bit (more generalized) from the notion of moral belief in a mentioned related work (Scherrer et al., 2023). The authors should clarify the terminology to avoid ambiguity or anthropomorphism.


2. The observed belief change appears closely related to well-established prompt-ordering and recency biases in language models. The work might be reframing standard contextual priming behavior (arising from attention over long sequences) as a safety concern specific to persistent LM agents. Clarifying this connection would improve the paper’s conceptual clarity. For example, can the observed belief change be explained by known recency bias? How much of the observed shift is due to recency bias (if any) versus other factors such as context density, model priors, or task type?




3. There’s a lack of in-depth or practical mitigation methods of the issue. A discussion on concrete solutions or robust evaluation of potential defenses may further enhance the significance of the work.

**Questions:**

Can you elaborate how the model behaves when the accumulated context contains contradictory or mixed arguments? For instance, if earlier context presents both pro-vegetarian and non-vegetarian positions, does the model’s conditional probability shift toward the most recent context? Or is it integrating both sides?

---

> ### Author Response · Authors · 2025-11-22
>
> We acknowledge the reviewer's recognition of our contributions that belief shift is a core risk of long-context LM behavior and our results show the divergence between stated beliefs and actual behaviors.
>
> > The paper appears to rebrand statistical shifts in output probabilities as ''belief change.'' Here, ''belief'' refers to the model’s response distribution conditioned on accumulated context, which seems to differ a bit (more generalized) from the notion of moral belief in a mentioned related work (Scherrer et al., 2023). The authors should clarify the terminology to avoid ambiguity or anthropomorphism.
>
> We thank the reviewer for pointing out the potential ambiguity. In our paper, "belief" simply refers to the model's response distribution conditioned on the context, meaning how the model tends to answer after receiving certain information. This use of "belief" does not imply any human-like belief or mental state. We clarified the terminology to avoid any impression of anthropomorphism.
>
> > Re: Weaknesses 2: Can the observed belief change be explained by known recency bias? How much of the observed shift is due to recency bias (if any) versus other factors such as context density, model priors, or task type?
>
> > Re: Questions: Can you elaborate how the model behaves when the accumulated context contains contradictory or mixed arguments? For instance, if earlier context presents both pro-vegetarian and non-vegetarian positions, does the model’s conditional probability shift toward the most recent context? Or is it integrating both sides?
>
> To better control and test whether the model follows only the most recent context or integrates earlier information, we conducted a **2-back** experiment before evaluation [1]. We evaluate on 3 progressive and 3 conservative readings across 10 survey topics, the results show clear integration rather than pure recency bias, with shifts moving partially but never overriding the influence of the original accumulated context.
>
> |          | conservative | progressive |
> |----------|--------------|-------------|
> | original | 90.0         | 70.0        |
> | 2-back   | 90.0         | 80.0        |
>
> [1] Is the N-Back Task a Valid Neuropsychological Measure for Assessing Working Memory? (2009)
>
>
> > There’s a lack of in-depth or practical mitigation methods of the issue.
>
> We agree that developing concrete mitigation strategies is an important next step, and we will add this point to the future direction section.

---

### Official Review · Reviewer_KZnD · 2025-10-31

**Soundness:** 2
**Presentation:** 3
**Contribution:** 1
**Rating:** 2
**Confidence:** 4

**Summary:**

The paper tests for changes in model self-expressed beliefs and actions when they engage in/undergo (i) debate, (ii) persuasion, (iii) information-seeking, and found large changes as a result of those interactions.

**Strengths:**

- The paper aims at a problem with growing importance in model deployment, that of context management, and subtle influence from the context information.
- Both self-reported beliefs and behavioral evidence (revealed beliefs) are accounted for in the experiments, which add to the soundness and generalizeability of the results.
- The paper is clearly structured and is easy to follow.

**Weaknesses:**

- Originality: Model under persuasion [1,2,3,4,5], model debate [6,7], and model reading [8], are all well-studied topics. The only setup not covered by these topics is belief change during deep research. However, the main finding in such a case is that such change exists and is large, which is very unsurprising (deep research was designed to make the model find new evidence that shifts its belief), and there are analogous results from evaluation on domains such as maths [9] and science [10].
    - Suggestion: The authors could consider going more applied. If the motivation is, as the title suggests, studying the side effects of context accumulation, then it's worth going beyond simple interactions modes like single dialogue sessions or single research sessions, but instead study e.g. real-world user interaction with chatbots that have long-term memory.
- Soundness: In section 4, the paper aims to find patterns of belief change. However:
    1. Table 2 and 4 show that different models have different levels of susceptibility to influence. However, no clear pattern arise other than that observation. Table 3 shows the choice of persuasion technique has an influence, but from the raw data it's hard to tell which techniques are more effective.
    2. Arguably the most interesting observation in the paper, "directionality of political belief shift", fails to account for many potential confounders, such as politicians from different camps tending to value political persuasion vs factual information to different degrees, or different levels of Internet availability of political speeches from different camps.

[1] Susceptibility to Influence of Large Language Models (2023)

[2] Towards Understanding Sycophancy in Language Models (2023)

[3] Moral Persuasion in Large Language Models: Evaluating Susceptibility and Ethical Alignment (2024)

[4] Extended AI Interactions Shape Sycophancy and Perspective Mimesis (2025)

[5] Persuade Me if You Can: A Framework for Evaluating Persuasion Effectiveness and Susceptibility Among Large Language Models (2025)

[6] Improving Factuality and Reasoning in Language Models through Multiagent Debate (2023)

[7] Debating with More Persuasive LLMs Leads to More Truthful Answers (2024)

[8] Belief Revision: The Adaptability of Large Language Models Reasoning (2025)

[9] WebThinker: Empowering Large Reasoning Models with Deep Research Capability (2025)

[10] Agentic Reasoning: A Streamlined Framework for Enhancing LLM Reasoning with Agentic Tools (2025)

**Questions:**

- Regarding, as mentioned on line 45, unreliability resulted from belief change due to context accumulation, I'd appreciate clarification on the exact scenario here: If all the context come from the user, or the target user distribution, then adapting to that preference doesn't seem to be a bad thing. Only if the drift is in opposite direction, in random direction, or overly large, does this seem concerning, but results seem to suggest belief change is in the same direction as user persuasion or external information, and there is also no comparison to a "rational belief update" baseline.
- Re "This shows that while stated belief stabilizes relatively early, longer conversations provide additional opportunities for models to adapt their actions, leading tomore pronounced behavioral change over time.": The trend seems unclear to me. Have you conducted statistical tests on the trend?

---

> ### Author Response · Authors · 2025-11-22
>
> We thank the reviewer for the constructive feedback and suggestions. However, we believe there is a misunderstanding of our paper. To clarify this, we (1) restate our main contributions, and (2) outline how our work differs from existing literature, including pointing out where these distinctions were discussed in the original submission.
> ## Originality
> > Model under persuasion [1,2,3,4,5], model debate [6,7], and model reading [8], are all well-studied topics.
>
> Thank you for providing these references. First, to clarify, our paper’s main point is to examine this research question (as noted in line 51 in our original submission):
> “In the context of persuasion/debate and research/reading, does accumulating context result in a belief shift in language models.”
>
> To organize these previous results that you cited and explain the differences, we created a table below of the previous work. We have added them to the related work in the revised paper (as noted in Section 5).
>
> | Paper  | Summary                                                                    | Persuasion/debate | Research/reading | Accumulating context | Belief shift |
> |--------|-----------------------------------------------------------------------------|--------------------|------------------|-----------------------|--------------|
> | [1]    | Studies whether LLMs can reliably simulate human behavioral and psychological effects of influence | Yes | No | No | No |
> | [2]    | Analyzes whether RLHF causes LLMs to favor agreeing with users instead of giving correct answers. | Yes | No | No | No |
> | [3]    | Tests whether LLMs can be persuaded in single interactions.                 | Yes | No | No | No |
> | [4]    | Examines whether models become more user-aligned after extended personal interaction history. | Yes | No | Yes | Yes |
> | [5]    | Measures persuasion success within those conversations                     | Yes | No | Yes | Yes |
> | [6]    | Studies whether debate improves reasoning accuracy.                         | Yes | No | Yes | No |
> | [7]    | Utilizes debate to improve the accuracy                                     | Yes | No | Yes | No |
> | [8]    | Examines appropriate belief updates in short, synthetic contexts.          | No  | No | No | Yes |
>
> As you can see, only [4][5] consider both accumulating context and belief shift, which is our core research question. So we would like to elaborate the novel contributions and findings in our paper with respect to this one:
> - We investigate how models' beliefs change as they read, conduct research, or engage in multi-turn interactions, rather than how they shift belief to align to a user’s preference.
> - Our key insight is that LM's stated beliefs and behavior can quietly drift, flip, or persist during routine interactions.
>
> &nbsp;
> &nbsp;
>
> > deep research was designed to make the model find new evidence that shifts its belief
>
> We don’t necessarily agree with this. Deep research is certainly designed to allow models to find new factual information and provide insights to the user (as cited, for instance on the OpenAI Deep Research announcement: https://openai.com/index/introducing-deep-research/). In addition, the papers [9] and [10] that you cited, state that the goal is to do better search, information mining, or reasoning capability. However, we have not found any document stating that deep research was designed to shift the model’s beliefs about the types of subjective or opinion-based questions that we handled in our research. In fact, there are significant efforts to prevent models from being unduly influenced in this way (such as OpenAI’s work on preventing sycophancy in gpt-4o: https://openai.com/index/sycophancy-in-gpt-4o/). If you have any references that state the contrary we would be happy to take a look at them and discuss them in detail.
>
> &nbsp;
> &nbsp;
>
> > it's worth going beyond simple interactions modes like single dialogue sessions or single research sessions, but instead study e.g. real-world user interaction with chatbots that have long-term memory.
>
> We would love to perform this experiment, but unfortunately do not have access to this variety of data, since data from sources such as ChatGPT is typically private. If there are public sources of data that we could use for this examination or realistic ways to do simulation we would be happy to take a look! But we also believe that the paper has merit without this additional study.

---

> > ### Author Response · Authors · 2025-11-22
> >
> > > Table 2 and 4 show that different models have different levels of susceptibility to influence. However, no clear pattern arise other than that observation.
> >
> > Thank you for pointing this out. Based on this comment we re-examined the description, and we agree that we could have been more insightful in our analysis of the results. Based on this, we have made several improvements to the re-submitted draft.
> >
> > **Table 2 & Table 3 (Table 6 in revised version)**
> >
> > The objective of Table 2 is to test whether persuasion techniques that are known to have varying degrees of influence on human listeners [11, 12, 13, 14, 15] can also influence LLMs. Interestingly, we find that it matters much less which persuasion technique is used, and more which underlying LLM is examined, and thus LLMs seem less sensitive to the particular persuasion technique that is used. We have clarified this in the draft. Due to the page limit, we move Table 3 (in original submission) to Appendix F (Table 6 in revised version).
> >
> > [11] Correcting inaccurate metaperceptions reduces Americans’ support for partisan violence (2022)
> >
> > [12] Evidence from Two Large Field Experiments that Peer Comparison Feedback Can Reduce Residential Energy Usage (2012)
> >
> > [13] Personalizing Moral Reframing in Interpersonal Conversation: A Field Experiment (2022)
> >
> > [14] Endorsements from Republican politicians can increase confidence in U.S. elections (2023)
> >
> > [15] Personal experiences bridge moral and political divides better than facts (2020)
> >
> > **Table 4 (Table 3 in revised version)**
> >
> > In our original submitted version, Table 4 (Table 3 in revised version) reports the directional shifts for non-intentional tasks, showing that models shift toward the ideology of reading materials (conservative vs. progressive).  In the revised version, we update the representation of Table 4 (Table 3 in revised version) to better visualize the directional patterns, including both the original stance of the model before reading or research, and after. We believe that this better demonstrates that the models tend to shift toward the source material's political orientation.
> >
> > > Arguably the most interesting observation in the paper, "directionality of political belief shift", fails to account for many potential confounders, such as politicians from different camps tending to value political persuasion vs factual information to different degrees, or different levels of Internet availability of political speeches from different camps.
> >
> > Thank you for pointing this out! We agree that there are potential confounders here, such as whether a particular reading material may have more emotional or factual language. Another confounder (which we tried to visualize better in the new version of Table ) is whether the LLM originally answers more questions with a conservative or progressive viewpoint. We have added additional discussion of this in Section 4.2 (lines 398-408) of the paper.
> >
> > ## Explanation of the Unreliability of Belief Shift
> > We understand the argument that belief shifts could be reasonable or even desirable. To clarify, we do not believe that belief shifts are uniformly good nor uniformly bad. Rather, we focus on characterizing the awareness and predictability of this shift phenomenon.
> >
> > Users interact with LM assistants over long-horizon sessions and often trust these systems. However, they may not be aware that the model's stated beliefs are gradually changing based on accumulated context. Prior studies show that models can shift their stance simply due to multi-turn user pressure [16], and that small, step-by-step changes can still accumulate into unintended drift even when each step looks reasonable [17]. These results suggest that the main issue is not the “direction” of the update, but whether the change is stable, predictable, and visible to the user.
> >
> > Even for non-safety or ambiguous topics, this kind of silent, unpredictable change can undermine user trust. This is why we consider "risk" an appropriate term: it highlights the potential reliability issue that comes from belief shifts happening without user awareness.
> >
> > [16] Measuring Sycophancy of Language Models in Multi-turn Dialogues (2025)
> >
> > [17] Auditing radicalization pathways on YouTube (2020)
> >
> > **Figure 2**
> > > Have you conducted statistical tests on the trend?
> >
> > We have updated the representation of Figure 2 in the revised version to make the initial shift from round 0 to round 2 more clear.  We observe that both information and empathy show a sharp increase from 0 to 2 rounds followed by a continued upward rise through 10 rounds, even though stated belief plateaus early. As the reviewer suggested, we performed a mixed-effects regression with round as a fixed effect and seed as a random effect, and found a significant positive effect of conversation length on behavioral shift (Information: $\beta$=0.010, *p*=0.039; Empathy: $\beta$=0.008, *p*=0.048), suggesting that longer conversations lead to larger behavioral changes.

---

### Official Review · Reviewer_VpF3 · 2025-11-02

**Soundness:** 2
**Presentation:** 3
**Contribution:** 2
**Rating:** 2
**Confidence:** 4

**Summary:**

The paper investigates how the stated "beliefs" of language models and their actions evolve as context accumulates over extended interactions, which includes either persuasive dialogue or neutral reading/research. Using a three-phase setup; (1) baseline evaluation, (2) exposure to new context, and (3) post-evaluation, the authors analyze four models (GPT-5, Claude-4-Sonnet, Grok-4, and Gemini). They find systematic and sometimes large belief shifts that are often also reflected in model actions, claiming that long-term contexts such as these can alter model outputs.

**Strengths:**

1. The setup of measuring both stated belief and action is nice

2. The task collection could serve as a benchmark for future work.

3. The setup is relatively simple and reproducible.

**Weaknesses:**

1. The evaluation appears to run each condition only once per model, without repetitions of the sampling process. All the reported models exhibit stochastic behavior. It is therefore unclear whether the reported belief shifts exceed the models' inherent stochastic variance. Without repeated baselines, we cannot tell if these are systematic updates or simply natural output fluctuations.

2. The authors frame belief change as a “risk,” but for non-safety-sensitive or open-ended topics, such change may actually indicate healthy adaptability. People also update their beliefs in light of new arguments. Right now, the framing risks overstating benign effects as safety concerns.

3. The moral and safety tasks were explicitly designed to produce disagreement across models (Sec. 3.1). This means there is no single “correct” stance, and therefore, a change in belief is not necessarily undesirable. In fact, shifts might reflect reasonable movement within an ambiguous space.

4. Not all four models are evaluated for all conditions, which weakens comparability.

**Questions:**

1. How do you differentiate a belief change from normal variability in model outputs? Did you repeat any baselines to estimate this?

2. How should we interpret belief changes in non-safety contexts? are they a robustness concern or an adaptive feature?

3. What if the presented context is just random text? Would the output changes be smaller?

nitpicks:
- Inconsistent use of “LM” vs. “LLM”
- Line 92: duplicated “by”

---

> ### Author Response · Authors · 2025-11-22
>
> We thank the reviewers for their detailed comments and suggestions.
> ## Weaknesses 1 & Questions 1
> > It is therefore unclear whether the reported belief shifts exceed the models' inherent stochastic variance. Without repeated baselines, we cannot tell if these are systematic updates or simply natural output fluctuations.
>
> >  How do you differentiate a belief change from normal variability in model outputs? Did you repeat any baselines to estimate this?
>
> For your concern about stochastic variance, we provide comprehensive statistical tests as Reviewer cgXV suggested (Appendix F, Table 5 in the revised version), showing that all observed belief shifts are statistically significant (p values range from < 0.0001 to 0.046) with effect sizes ($\eta^2_p$) ranging from 0.0017 to 0.4905. We use these repeated-measures ANOVAs to show that the belief shifts in our study are systematic rather than random fluctuations.
> Additionally, we would like to clarify that each LLM query is an independent generation process with its own output distribution. Running additional seeds would reduce sampling variance but would not change our main conclusion given that our statistical tests already demonstrate highly significant effects.
> - For intentional tasks, we evaluate (30 safety queries + 30 moral dilemmas) × 3 seeds = 180 independent samples per model.
> - For non-intentional tasks, we originally evaluated 39 political questions × 14 reading topics = 546 samples. In the revised version, we added 12 strong progressive survey questions to balance the initial belief distribution, expanding to 51 questions × 14 topics = 714 samples per model.
>
> The fact that these result in statistically significant shifts demonstrates that the existing sample size is sufficient to reliably detect the systematic belief shifts we report in our paper.
>
>  &nbsp;
>  &nbsp;
>  &nbsp;
>  &nbsp;
>
> ## Imprecise Usage of Term "Risk" (Re: Weaknesses 2 & 3; Questions 2)
> Thank you for pointing this out. Our core stance with respect to this is that belief shift is not necessarily a bad thing, but at the very least users of systems should be aware that there is a possibility for the models’ beliefs to shift. Thus the risk is more in the silent change of beliefs without user awareness, as opposed to belief shifts being inherently a bad thing. We re-read the paper and think that the current statements are appropriately qualified, but if you have alternative suggested wording we would be happy to consider it!
>  &nbsp;
>  &nbsp;
>  &nbsp;
> ## Weaknesses 4
> > Not all four models are evaluated for all conditions, which weakens comparability.
>
> We agree with this concern! To be honest, this was originally due to financial constraints (the experiments come with considerable cost), but we agree that having all of the models would be more informative, so we added the additional experiments. In the revised version we ran the same set of models on all tasks, but were not able to do 3 seeds for the more expensive models. You can see the revised models in Appendix G.
>
>  &nbsp;
>  &nbsp;
>  &nbsp;
>  &nbsp;
>
> ## Questions 3
> >  What if the presented context is just random text? Would the output changes be smaller?
>
> We thank the reviewer for raising this important question. The core of this question is whether belief shifts are driven by specific topic-relevant information or by long-horizon context accumulation. To address this, we provide an *embedding analysis* on in-depth reading in the revised version (Table 4) to show that belief shifts are not caused by topic-relevant information in the reading materials.
>
> We embed each sentence in the reading materials and each political survey topic. We then use cosine similarity to identify the top-k (k=10, 50, 100, 200) most semantically relevant sentences. We investigate two conditions: (1) mask the top-k most relevant sentences with [MASK] tokens, (2) concatenate the top-k relevant sentences as reading materials. We select 10 political survey topics for evaluation and run this analysis on 3 conservative and 3 progressive topics that show the largest shifts, using Grok-4 and DeepSeek-V3.1.
>
> Table 4 (in revised version) shows that masking topic-relevant sentences or using only topic-relevant sentences does not eliminate shifts. This demonstrates that belief shifts primarily come from accumulated context, rather than from specific topic-related information.
>
>  &nbsp;
>  &nbsp;
> ## nitpicks
>
> We thank the reviewers for their careful reading, and we have addressed all noted nitpicks in our revised submission.

---

### Official Review · Reviewer_cgXV · 2025-11-05

**Soundness:** 2
**Presentation:** 3
**Contribution:** 3
**Rating:** 6
**Confidence:** 3

**Summary:**

This paper attempts to investigate whether LLMs change their internal beliefs after receiving a long context input. The authors conducted a simple experiment with three stages; stage 1 is a pre-examination stage for LLM's initial belief, stage 2 is an intervention forcing LLMs' intentional (persuasion and debate), and non-intentional (reading and research) shifts, and stage 3 is a post-examination stage for LLM's next belief. The authors tested four LLMs (GPT-4 and Claude 4 Sonnet for both shifts; and Gemini-2.5 Pro and Grok-4 for non-intentional shifts only) and analyzed the differences between settings. As a result of experiment, they found that LLMs are affected by long context input, especially by intentional ones. Also, they found a misalignment between stated beliefs and actual behavior of LLMs, and the effect of length (whose size and shifting patterns might be different due to shifting types). And, the paper discusses several differences that each shifting techniques produce in LLMs' belief statements or behaviors.

**Strengths:**

- Designing an experimental framework to measure the effect of intentional/non-intentional shifts which can possibly introduced by a long context input
- Sufficient analysis that possibly uncovers some pathways how intentional/non-intentional shifts affect LLMs' belief.

**Weaknesses:**

- Despite the depth of analysis, the experimental setup seems not systematic. As the current condition might introduce several unwanted confounding effects, the result analysis cannot be fully attributed to the experimental conditions (i.e., designed shifts). See Question A.

**Questions:**

## Question A. Experimental setup.

A1. Statistical testing. I'm asking the following changes because the authors already used ANOVA in Section 4. I think the authors already have sufficient knowledge about the statistical testing. And I believe that these changes can strengthen the paper's finding.

A1-1. The experiment follows a general pattern of intervention-based experiment: pre, intervention, and post. This is a repeated measurement setting, which requires paired testing. But the current result does not provide such result; instead, the result currently provides only the mean differences. As randomness (due to stochastic sampling in generation) produces statistical deviations of the result, the difference itself cannot conclude the difference actually exists or not. Thus, I recommend use repeated measurement methods for ANOVA to avoid such errors.

A1-2. Also, this paper uses two-factor setting: intervention (shift) and model types. Thus, using an ANOVA might be inappropriate because ANOVA does not assumes such two-factor setting. Please consider ART ANOVA or other methods designed for both repeated measurement and multiple factors.

A2. Model selection. Why did the authors use different models for intentional and non-intentional shifts? Because of the difference, it seems that the range of analysis is limited. What happens if we test Grok and Gemini on intentional shifts?

A3. Shifting methods.

A3-1. The authors mentioned that they used different topics in shifting methods. As LLMs are easily affected by topics or inputs (as the authors already know), the difference in topic might influence the result. It seems that the topics used in intentional shift can be used for non-intentional and vice versa. Is there any reason behind this decision? Does the difference affect the result?

A3-2. For the intentional shift, the authors decided to use a famous book written by Michael Sandel. Is its content possibly learned by LLMs during the pretraining?

A3-3. For the unintentional shift, the authors decided to provide a reading material related to the task. Does the material provide information for/against LLMs' belief? I'm asking this because LLMs do have confirmation bias and other kind of cognitive biases, which can affect the decision or belief of LLMs.

---

> ### Author Response · Authors · 2025-11-22
>
> We thank the reviewer for acknowledging our in-depth analysis and for the constructive suggestions. In response, we have expanded our statistical analysis and model comparisons to address each concern directly. We provide detailed clarifications below.
>
> ## Model Selection
> > Why did the authors use different models for intentional and non-intentional shifts?
>
> Our initial model selection was driven by two factors: 1) computational cost constraints, and 2) the non-intentional shift phenomenon was particularly surprising, since the belief shifts from seemingly neutral information processing (reading and research) are more unexpected than shifts from direct persuasion. Therefore, we prioritized testing additional models (Grok-4 and Gemini-2.5-Pro) on non-intentional tasks.
> In the revised version, we provide more comprehensive comparisons, we add two open-source models (GPT-OSS-120B and DeepSeek-V3.1) across all tasks in the revised version. We show the results in blue in Table 1. These open-source models exhibit shifts in both intentional and non-intentional tasks, with generally smaller magnitudes than closed-source models. This supports our finding that belief malleability is not limited to specific model families.
>
> > What happens if we test Grok and Gemini on intentional shifts?
>
> We also tested Grok-4 and Gemini-2.5-Pro on intentional tasks to directly address the reviewer's question. Due to cost constraints, we ran one seed for this exploratory analysis. Table 7 (Revised version) shows that Grok-4 exhibits large shifts in both debate (68.4%) and persuasion (65%), while Gemini shows smaller shifts (17.9% and 35.9%). The repeated-measures ANOVA demonstrates Grok-4's shifts are highly significant (debate: p=3.45e-05 and persuasion: p=1.91e-05), and Gemini shows significant shift in persuasion (p=0.046) but not debate (p=0.47), possibly due to the limited number of examples we used. The ART ANOVA shows no significant difference between the two models (p>0.25 for both tasks).
>
> These additional results align with and support our main conclusion: LM assistants exhibit systematic belief changes under context accumulation, with the degree of vulnerability varying across models but the phenomenon being consistent across different architectures.
>
> ## Systematic Experimental Setup
> > I recommend use repeated measurement methods for ANOVA to avoid such errors.
>
> We agree that our 3-stage (pre-intervention-post) experiment design requires repeated-measures analysis. We have conducted repeated-measures ANOVA for both intentional and non-intentional shift tasks. For intentional tasks, closed-source models show significant shifts in both debate (GPT-5: p < 0.0001, $\eta^2_p$ = 0.4905; Claude-4: p = 0.0470, $\eta^2_p$ = 0.0260) and persuasion (GPT-5: p < 0.0001, $\eta^2_p$ = 0.4247; Claude-4: p = 0.0065, $\eta^2_p$ = 0.0107). We also observe significant shifts with smaller effect sizes ($\eta^2_p$ = 0.0017 to 0.0801) in open-source models. For non-intentional tasks, all models show highly significant shifts (all p < 0.0001), with consistently larger effect sizes in reading tasks ($\eta^2_p$ = 0.0684 to 0.2112) compared to research tasks ($\eta^2_p$ = 0.1036 to 0.1948). This pattern suggests that prolonged, coherent context exposure produces stronger belief changes than fragmented information gathering.
>
> > Please consider ART ANOVA or other methods designed for both repeated measurement and multiple factors.
>
> We also computed Aligned Rank Transform (ART) ANOVA to test model differences. For intentional tasks, we separately calculated ART ANOVA for the two model pairs since they use different data. This is because our experimental design requires LMs to hold counter positions on each moral or safety query, so we can only compare models within the same group. Within closed-source models (GPT-5 vs Claude-4), the ART ANOVA shows these two models differ significantly in their shifts (F=5.3, p=0.0229). Within open-source models (GPT-OSS-120B vs DeepSeek-V3.1), the difference is even stronger (F=31.0, p<0.0001). For non-intentional tasks, we use the same data across all models, allowing direct comparison. The ART ANOVA shows a significant main effect (F=17.0, p<0.0001), indicating that models differ systematically in their responses to reading and research tasks.
>
> These statistical tests further prove that the stated belief shifts reported in Table 1 are statistically significant and robust, supporting our main finding that LM assistants exhibit systematic belief changes as context accumulates. We have added the complete statistical results in the revised paper as Appendix F, Table 5.

---

> ### Author Response · Authors · 2025-11-22
>
> ## Shifting Methods
> > It seems that the topics used in intentional shift can be used for non-intentional and vice versa. Is there any reason behind this decision? Does the difference affect the result?
>
> We use different topics for intentional and non-intentional tasks because we have different experimental assumptions:
> For intentional tasks, we require two models to hold counter positions on each topic. We tested political survey topics but found all LMs consistently favored progressive positions, making this analysis difficult. Moral dilemmas and safety queries naturally elicit diverse responses, allowing us to pair the LM assistants with opposing stances.
> For non-intentional tasks, we need topics with clear directionality to observe systematic belief shifts. Political topics have a well-defined conservative-progressive dimension, allowing us to examine whether models shift toward the political orientation of the reading materials. As shown in Table 3 in the revised version, models consistently exhibit directional shifts that align with the source material's ideology. Moral dilemmas and safety questions lack this dimensional structure, making it more difficult to measure directional changes.
> We describe this justification further in the revised version (see Experimental Setup, Models and datasets paragraph).
>
>
> > For the intentional shift, the authors decided to use a famous book written by Michael Sandel. Is its content possibly learned by LLMs during the pretraining?
>
> We agree that Michael Sandel's work may exist in the pretraining data. However, to clarify, in-depth reading is not about learning new knowledge. Reading familiar content still constitutes context accumulation through prolonged engagement. This is similar to how humans may shift their views when re-encountering arguments. We define "in-depth reading" in our paper as the LM assistant passively reading curated documents on given topics, with these materials forming the accumulated context for the model. In our study, what matters is the accumulated context from extended exposure, not the novelty of the information.
>
>
> > Does the material provide information for/against LLMs' belief? I'm asking this because LLMs do have confirmation bias and other kind of cognitive biases, which can affect the decision or belief of LLMs.
>
> We agree that the directionality of reading materials is an important factor. Our reading materials include both conservative and progressive perspectives (7 conservative topics and 7 progressive topics). To provide more comprehensive evaluation, we added 12 progressive survey questions with somewhat more radical stances to the original 39 political survey questions in the revised version to better balance the initial belief distribution of the LM assistants. We also updated the representation of Table 3 in the revised version (as noted in Table 4 of the original submission) to clearly show the directionality of belief shifts. Our results show that LM assistants shift toward the ideology of the reading materials. Reading conservative materials produces larger shifts than reading progressive materials because models still initially lean progressive on most questions. This directional analysis suggests that the direction of reading materials matters for belief shifts.

---

### Meta-Review · Area_Chair_RWjU · 2026-01-07

**Summary:**

This paper examines the changes of the belief of language models under long context input in form of model debate, persuasion, or reading. Reviewers had several concerns about the paper, including the fact that experiments were not systematic and did not control for confounders, the results were not surprising, and that there were no consistent takeaways from the paper. In addition, there were concerns that the results of the paper followed well-known trends in LLMs of recency bias and prompt ordering. Given these outstanding concerns from all reviewers, the AC believes the paper is not ready for publication in its current form.

**Reviewer Concerns:**

Reviewers had concerns over the fact experiments were not systematic and did not control for confounders, the results were not surprising, and that there were no consistent takeaways from the paper. In addition, there were concerns that the results of the paper followed well-known trends in LLMs of recency bias and prompt ordering. In response to the reviewer's concerns, the authors ran additional tests with statistical significance, which partially resolves these issues but not fully.

**Reviewer Scores:**

I believe that major concerns of reviewers  dTDQ,  KZnD, and VpF3 were not well resolved and they would have maintained their negative review.

---

### Decision · Program_Chairs · 2026-01-26

Reject